# Response strategies to acute and chronic environmental stress in the arctic breeding Lapland longspur (*Calcarius lapponicus*)

Zhou Wu [1] ✉, Matthew M. Hindle[1], Valerie R. Bishop[1], Angus M. A. Reid[1], Katarzyna Miedzinska[1], Jonathan H. Pérez[2,3], Jesse S. Krause[4], John C. Wingfield[2], Simone L. Meddle [1] & Jacqueline Smith [1]

The potentially devastating effects of climate change have raised awareness of the need to understand how the biology of wild animals is influenced by extreme-weather events. We investigate how a wild arctic-breeding bird, the Lapland longspur (*Calcarius lapponicus*), responds to different environmental perturbations and its coping strategies. We explore the transcriptomic response to environmental adversity during the transition from arrival at the breeding grounds to incubation on the Arctic tundra. The effects of an extremely cold spring on arrival and a severe storm during incubation are examined through RNA-seq analysis of pertinent tissues sampled across the breeding cycle. The stress response, circadian rhythms, reproduction, and metabolism are all affected. A key gene of the Hypothalamic-Pituitary-Adrenal axis, *FKBP5*, was significantly up-regulated in hypothalamus. The genome assembly and gene expression profiles provide comprehensive resources for future studies. Our findings on different coping strategies to chronic and acute stressors will contribute to understanding the interplay between changing environments and genomic regulation.

## Main

Wild-living species are often exposed to challenging environmental conditions, with many of these situations arising from extreme weather events[1,2]. Detrimental weather conditions such as variations in temperature, precipitation, and snow cover (e.g., https://www.worldclim.org/), can have a prevailing impact on the physiology and behaviour of wild-living animals. The response to these events is being influenced by the interplay between the type of climatic challenges and the life history stage during which they are experienced (e.g., breeding, winter, or migration)[3]. For instance, a chronic event constraining food availability prior to the time of arrival can result in an unpredictable adverse situation and aberration in phenology, whereas a storm event during egg-lay and incubation period can force birds to face the dilemma of either allocating energy for thermoregulation or conserving resources for reproduction[4,5].

Inclement weather events impose negative energy challenges, and birds have been observed to prioritise survival over other life history activities (e.g., breeding)[6,7]. Such environmental perturbations influence many aspects of the biological process, with the endocrine system acting as a crucial signalling mediator to regulate biological responses like energy metabolism and reproduction[4]. From an energy metabolism perspective, extreme climate events impose a physiological challenge for survival (e.g., food and water deprivation) due to the overall energetic supply failing to meet the overall demand to maintain regular life cycle activities, which is described as Type I allostatic overload[3,8]. The concept of allostasis describes how animals 'maintain stability through change'[9], which accounts for both predictable and unpredictable changes.

In response to the state of allostatic load, animals may move into an emergency life history stage (ELHS) to reduce energy costs, deviating from the normal life history stages; for example, suspension of breeding activity[3]. Essentially, animals redistribute their available resources and internal energy to promote survival. During this process, the Hypothalamic-Pituitary-Adrenal (HPA) axis is responsible for regulating the physiological response by elevating a cascade of hormonal secretions to cope with environmental perturbations[3], with cortisol/corticosterone being the most

[1]The Roslin Institute and Royal (Dick) School of Veterinary Studies R(D)SVS, University of Edinburgh, Easter Bush Campus, Midlothian, UK. [2]Department of Neurobiology, Physiology and Behavior, University of California, Davis, CA, USA. [3]Department of Biology, University of South Alabama, Mobile, AL, USA. [4]Department of Biology, University of Nevada Reno, Reno, NV, USA. ✉e-mail: zhou.wu@roslin.ed.ac.uk

important glucocorticoids[10]. Different types of environmental perturbation can cause distinct allostatic load, resulting in various coping strategies dependent on whether the stressor is acute or chronic, and when it occurs according to life history stage. These factors collectively determine how animals pursue the optimal strategy, thereby altering their current physiological and behavioural status. The response strategies and outcomes are often mediated by a change in gene expression. Therefore, transcriptomic studies on how gene expression changes in response to inclement weather events can improve understanding of the genomic basis of these coping strategies.

In order to understand the different survival mechanisms and underlying physiological regulation, we investigated transcriptomic differences in free-living Lapland longspurs (*Calcarius lapponicus*) in response to two types of inclement weather conditions, which occurred across different life cycle stages. We examined samples collected at the arrival stage during an extremely cold spring, resulting in delayed Lapland longspur arrival and clutch initiation, and from the incubation period, which coincided with a heavy snowstorm, where birds abandoned nesting behaviours and entered an ELHS.

The Lapland longspur is an Arctic breeding songbird, known to experience environmental adversity during its life cycle (Fig. 1)[7,11]. The spring in Alaska, USA in 2013 was documented as extreme, having higher snow cover compared to all other studied years (i.e., 2010 to 2014)[4] and resulting in limited food resources[5,6]. Previous studies showed that increased snow cover and low temperature in the extremely cold spring of 2013 significantly influenced their breeding behaviour, including delayed arrival and delayed clutch initiation by ~3 to 10 days[4]. This delay in breeding-cycle phenology can have significant influence on the survival of the birds and their offspring because of the potential consequences of phenological mismatches[12]. In comparison, during an acute snowstorm in 2016, birds abandoned their nests, and exhibited elevated corticosterone levels[5,7]. We have previously shown that Lapland longspurs experienced elevated concentrations of corticosterone, indicative of a physiological response to environmental challenge, and suggestive of entry into the ELHS[5,7,10].

To study the coping strategies associated with ELHS activation in Lapland longspurs following severe inclement weather, it is important to examine gene expression changes. To ensure accurate quantification of gene expression, we developed a high-quality chromosome-level Lapland longspur genome assembly, and RNA sequencing dataset collected across different tissues. To assess overall stress physiology, based on previous knowledge of the physiological changes in extreme weather events,

including affected reproductive behaviour, hematocrit levels, HPA axis activity, and body condition[5,7,11], we examined the testis, heart, hypothalamus and liver to identify differentially expressed genes in three environmental or life history comparisons, namely (i) birds arriving in their breeding grounds during an extreme spring (May 2013) compared to a normal spring arrival (May 2016); (ii) birds incubating during a snowstorm (5th June 2016) compared with storm-free incubation (17th June 2016), for which we further examined the pituitary gland, adrenal gland, and fat tissue; (iii) birds at different annual life-cycle stages - incubation versus arrival at the breeding ground, in storm-free conditions.

## Results

### Assessment of genome assembly

High-quality long read sequence was included for contig construction and Omni-C reads (Fig. S1A, Supplementary material 1) for scaffolding[13]. We present an assembly representing a total size of 1,154,813,007 bp, scaffold count of 457, and scaffold N50 of 72.36 Mb (Table 1, Figs. S2 and 2A). Results of repetitive element identification and BUSCO assessment indicate that the Lapland longspur genome is comparable to other birds of the *Fringillidae* family[14] which includes closely related and well-studied species. Total repeat content of Lapland longspur is 13.45%, comprising 4.28% LINEs and 0.03% SINEs (Supplementary Table S1 and Fig. S1B), which is also close to that of other *Fringillidae* birds, with these elements ranging from 8.5 to 15.4%, 3.92 to 4.02% and 0.06 to 0.08%, respectively[14]. Similarly, the Lapland longspur genome shows good completeness with BUSCO assessment (aves_odb10: C:96.1% [S:92.5%, D:3.6%], F:0.8%, M:3.1%, n:8338) (Fig. 2B), which is similar to that of previously assembled passerine birds[14]. In addition, we aligned short-read sequences of an individual to validate the assembly, which resulted in a mapping rate of 98.96% with a mean mapping quality across the genome of 43.81 and a general error rate of 0.0191.

### Comparative genomic analysis with zebra finch

We aligned our Lapland longspur assembly to the reference genome of the zebra finch (*Taeniopygia guttata*, bTaeGut1.4.pri). Macro-chromosomes, intermediate chromosomes and known micro-chromosomes are represented by Scaffolds 1 to 23, ordered by chromosome size, as shown in Fig. 2C. Chromosome assignment was based on the zebra finch genome, including the Z chromosome, and chromosomes 1A and 4A (Fig. S1C). Acknowledging the alignment uncertainty and potential scaffolding errors for some of the micro-chromosomes, we performed the comparative

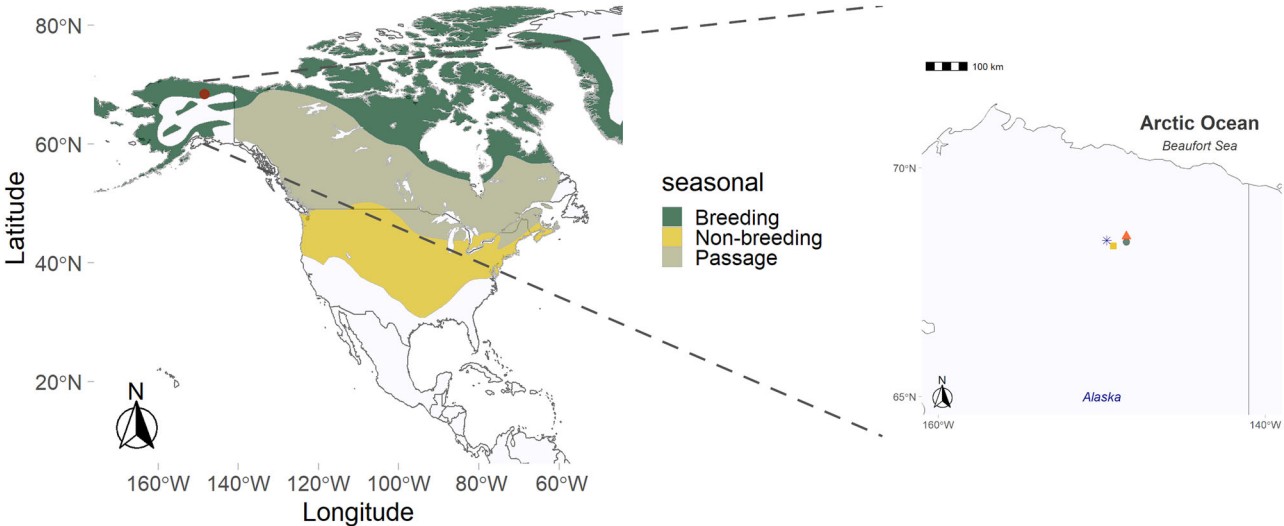

**Fig. 1 | Map of Lapland longspur distribution and sample study sites.** The left panel shows the seasonal distribution of the Lapland longspur. The right panel shows the four field study sites based in Alaska, USA. Samples were collected at these sites which included Toolik Field Station (TLFS) shown by a green circle, Galbraith Lake shown by a yellow rectangle, Pump Station 3 by a red triangle, and MP 297 as a blue asterisk.

analyses for macro-, micro- and sex chromosomes separately (Fig. S1D). Because of this, the submitted assembly on GenBank (GCA_039654755.1) includes chromosomes with the highest confidence of assignment as well as the MT sequences (CM077482.1).

Assembly of the mitochondrial (MT) genome resulted in a 16.827 kb DNA sequence with a circular formation (Fig. S3), whose length is comparable to that of other species in the *Fringillidae* family[15]. When the Lapland longspur MT genome was aligned to the MT sequence of chicken, Red Crossbill (*Loxia curvirostra*)[16] and Japanese Grosbeak (*Eophona personata*)[15], with the latter two being closely-related species from the *Fringillidae* family, we found a linear alignment between the Lapland longspur MT sequence and reference sequences, which suggests our MT assembly and the proximal starting position exhibit a typical avian mitochondrial structure.

## Gene expression profile

In order to understand the gene expression changes occurring in response to inclement challenges, we generated data from 48 RNA-seq libraries representing different tissues, different environmental conditions, and different stages of the breeding cycle. Subsequently, we performed a set of focussed RNA sequencing on an additional 18 samples representing three tissues of HPA importance during incubation in both snowstorm and normal conditions.

Overall, the RNA-seq dataset showed good mapping quality to the current genome assembly, with an average unique mapping rate of 75.42% and an average number of uniquely mapped reads of 40.7 million reads per sample. After assigning the reads to individual genetic features, an average of 56.7% reads were confidently assigned, with 4.75% reads unassigned and 33% multi-mappers (Fig. S4A). The overall expression profile displays a tissue-specific pattern separated based on their tissue type and clustered closely within the same tissue, independent of weather conditions and life history stage, which suggested an appropriate sampling process (Fig. S4B).

Our previous study showed that birds arriving during the 2013 extreme spring had significantly lower body mass, fat scores, and haematocrit levels compared to those arriving under benign conditions[11]. Size and volume of cloacal protuberance (CP) can also indicate the reproductive state of male birds[17]. Measurements of phenotypes and plasma corticosterone levels adapted from previously published studies[7,11] are available in Fig. S5.

**Table 1 | Assessment of the Lapland longspur (*Calcarius lapponicus*) genome assembly**

| Lapland longspur genome | Summary |
|---|---|
| Total Length (bp) | 1,15,48,13,007 |
| Counts of scaffold sequences | 457 |
| Largest scaffold length | 15,78,14,153 |
| Scaffold N50 | 7,23,67,164 |
| Counts of N50 | 6 |
| Scaffold N90 | 1,44,54,378 |
| Counts of N90 | 22 |
| GC content (%) | 42.72% |
| N Length | 1,06,522 |
| N content (%) | 0.009% |
| Counts of contigs | 1,492 |
| Maximum length of contigs | 1,01,35,631 |
| ContigN50 | 19,95,792 |
| Counts of contig N50 | 174 |
| Contig N90 | 4,27,527 |
| Counts of contig N90 | 625 |

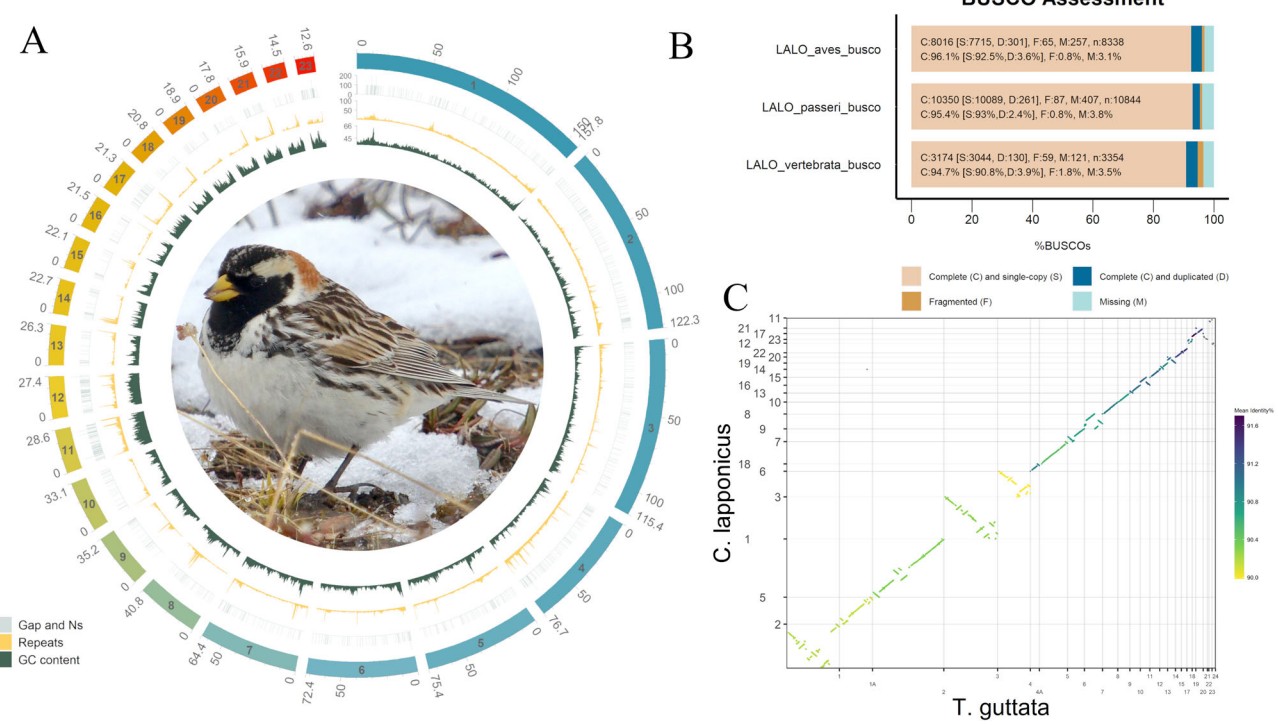

**Fig. 2 | Genome assembly of Lapland longspur (*C. lapponicus*). A** Circos plot showing, from outer to inner circles, the primary scaffolds (1-23), the gaps and Ns (in 50 kb windows), repetitive elements (200 kb windows), and GC content (200 kb window) in the genome. **B** BUSCO assessment of the genome assembly using three databases (*aves*, *vertebrata*, and *passeriformes*, version odb10). Different colours in the bar chart represent the number and percentage of genes of different features. **C** Dot plot shows the alignment of Lapland longspur scaffolds with the genome of zebra finch (*T. guttata*); the colour of segments represents the mean identity of the alignment.

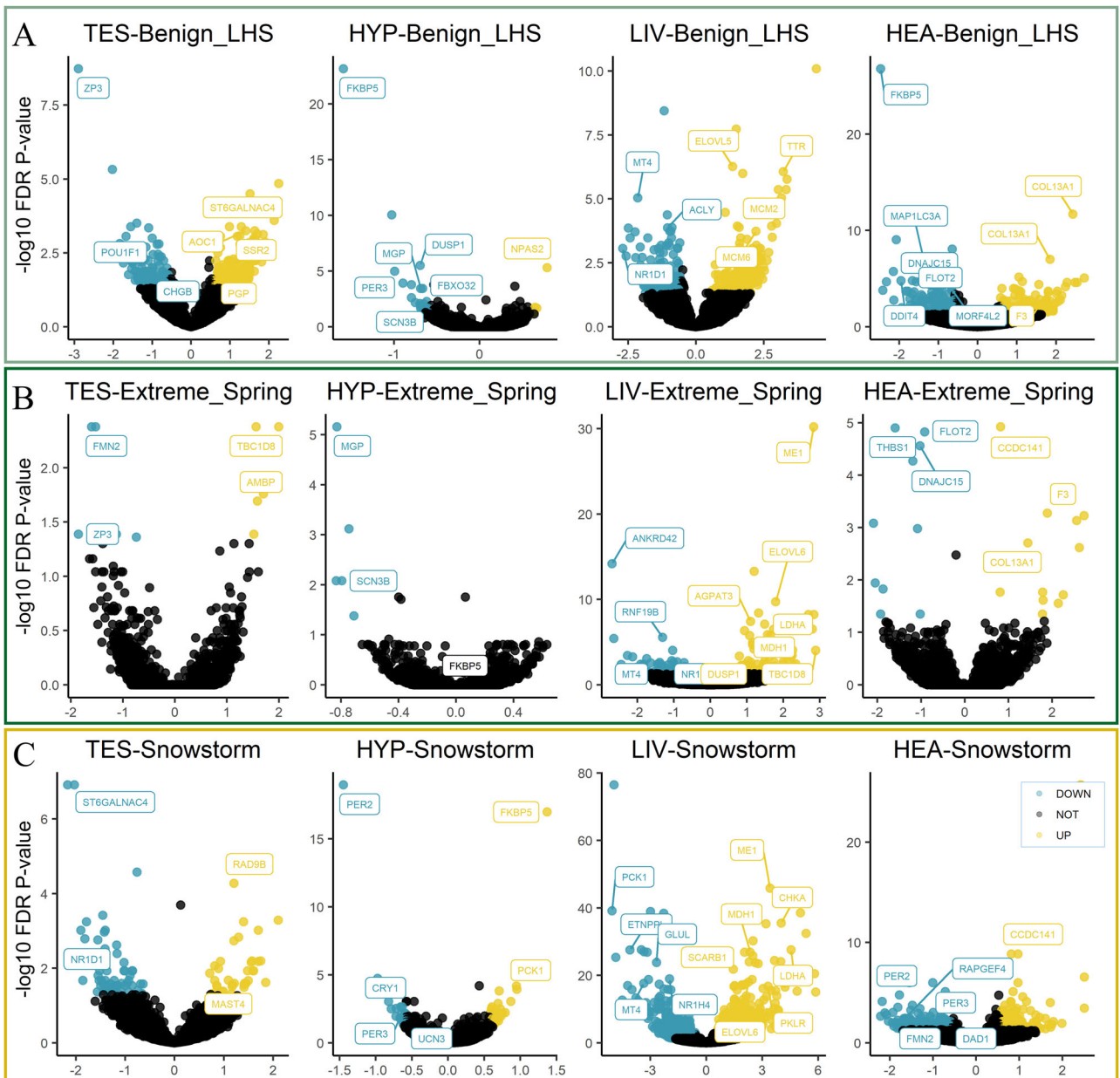

**Fig. 3 | Differentially expressed genes (DEGs) identified for all comparisons.**
**A** Volcano plot shows DEGs detected in birds during incubation compared to arrival under benign weather conditions (Benign_LHS). **B** DEGs identified in birds arriving in extreme spring compared to a normal spring (Extreme_Spring). **C** DEGs detected in birds incubating during a severe storm compared to normal incubation (Snowstorm). For each comparison, four tissues are presented: testes (TES), hypothalamus (HYP), liver (LIV), heart (HEA). Yellow and blue dots represent significantly up- or down-regulated genes respectively compared to the control conditions (threshold: FC > 1.5, FDR $P$ < 0.05).

## Genes differentially expressed between life history stages under normal conditions

We evaluated how tissues change their gene expression pattern between life history stages (arrival and incubation) when there is no specific perturbation. This focused on transition from pre-breeding to breeding status, shedding light into tissue-specific expression dynamics in relation to the reproductive phase. Differentially expressed genes (DEGs) were identified by comparing birds during incubation with birds arriving at the breeding ground (Condition 3 v 2, as described in Methods) (Fig. 3A).

In testicular tissue, we discovered 383 up- and 160 down-regulated DEGs (Supplementary Table S2), with the most significant DEG being the gene encoding zona pellucida glycoprotein 3 (*ZP3*), which shows reduced

expression during incubation compared to arrival (FDR adjusted $P = 1.87 \times 10^{-9}$ and fold change = 7.46). The expression of *ZP3* has recently been reported in the process of spermatogenesis in human and mouse testes[18]. Interestingly, transcription factor nuclear factor-κB (NF-κB) is seen to be activated (Fig. S6A), supporting regulated expression of many genes involved in carbohydrate metabolism and immunity[19].

The hypothalamus showed 4 up- and 22 down-regulated DEGs. *FKBP5*, the gene encoding FK506 Binding Protein 5, showed a reduced expression pattern (FDR adjusted $P = 7.02 \times 10^{-24}$, fold change = 3.02) when birds entered the incubation stage. This was also observed in the liver and heart tissues. This reflects the complex regulation between FKBP5 and glucocorticoid receptor (GR) in multiple tissues. Results also showed that

*DUSP1*, which is important for glucocorticoid-driven anti-inflammatory function, exhibited reduced expression in the hypothalamus of incubating birds. *FBXO32*, which regulates muscle atrophy, was seen to be up-regulated in the hypothalamus upon arrival at the breeding ground. Another gene, *PER3*, which is involved in circadian timing, was differentially expressed in heart and liver. DEGs detected in the heart (195 up- and 193 down-regulated) were, unsurprisingly, most significantly enriched for "cardiovascular system development and function" (Fig. S6B).

In the liver (231 up- and 118 down-regulated DEGs), the gene encoding proto-oncogene tyrosine-protein kinase (*KIT*) exhibited a 21.58-fold change in expression level (FDR adjusted $P = 8.18 \times 10^{-11}$). KIT is involved in many important KEGG pathways, for instance the MAPK and Ras signalling pathways[20]. Additionally, increased expression of Elongation Of Very Long chain fatty acids protein 5 (*ELOVL5*) during incubation suggests fatty acid synthesis activity was accordingly regulated. Members of the Mini-Chromosome Maintenance proteins (MCM) gene family, including *MCM2* to *MCM6*, were more highly expressed during incubation compared to arrival. MCM genes form an MCM complex that precedes DNA replication[21] which is supported by significantly enriched GO terms, such as "DNA replication" ($P = 9.4 \times 10^{-6}$) and "cell cycle" ($P = 1.5 \times 10^{-8}$). Furthermore, genes encoding Heat Shock Protein (HSP) family members or associated proteins were differentially expressed. For instance, the expression of HSPB1 associated protein 1 (*HSPBAP1*) is down-regulated (FDR adjusted $P = 0.02$, fold change = 3.79) in liver, while the expression of Heat Shock Protein Family B (Small) Member 9 (*HSPB9*) showed a down-regulated pattern in heart (FDR adjusted $P = 2.60 \times 10^{-4}$, fold change = 2.04).

We discovered the gene encoding Pyruvate Dehydrogenase Kinase 4 (*PDK4*) to be differentially expressed in all tissues tested; this gene is shown to reduce its expression level when entering into the incubation stage. PDK4 is located in the mitochondrial matrix and plays a role in regulation of glucose metabolism by inhibiting the pyruvate dehydrogenase which catalyses pyruvate conversion to acetyl-coenzyme A[22]. When comparing the canonical pathways among tissues, (excluding hypothalamus because of the low number of DEGs), we found very few common pathways with similar activation pattern, but pathways associated with cell cycle were prominently overrepresented (Fig. S6C). We also found significant overrepresentation of the E2F transcription factor family (Fig. S6D), including *E2F1*, *E2F1DP1RB* and *E2F4DP1* which are associated with regulation of genes involved in cell cycle progression.

### Differentially expressed genes associated with extreme spring
Focussing on DEGs detected in response to a chronic stressor, we investigated samples from birds that had experienced an extremely cold spring upon their arrival in the breeding grounds and compared them to birds arriving under benign conditions (Condition 1 v 2, as described in Methods) (Fig. 3B). In line with the curtailed reproductive activity that was observed in the field, we found interesting differential expression of genes in the testes. Although the number of DEGs was not shown to be large (5 up- and 5 down-regulated), their functional association with reproduction was apparent. In particular, genes encoding ZP3 and Formin 2 (*FMN2*) showed significant down-regulation. Recent studies identified testicular *ZP3* expression during spermatogenesis[18,23], which further supports our observation of its reduced expression during normal transition between breeding stages. This suggests the molecular mechanism underlying decreased reproductive activity may have its roots in gametogenesis[24,25]. Enriched biological functions associated with DEGs included "cell morphology", "organ morphology" and "reproductive system development and function" (Fig. S7A).

In keeping with our observation of reduced body mass and fat stores[11], we found genes involved in glycolysis/gluconeogenesis, tricarboxylic acid cycle and fatty acid biosynthetic process to be highly expressed in the liver. We found 95 up- and 67 down-regulated genes in the liver. Among those, a significant number of DEGs are involved in "lipid metabolism", "molecular transport", and "carbohydrate metabolism". This includes genes such as

Malic Enzyme 1 (*ME1*), Triokinase and FMN Cyclase (*TKFC*), Pyruvate Dehydrogenase E1 Subunit Alpha 2 (*PDHA2*), and Citrate Synthase (*CS*). Pathways associated with the tricarboxylic acid (TCA) cycle and respiratory electron transport show the most interesting changes in liver, with significantly positive Z-scores (Fig. S7B). The most significant pathway is the extrinsic prothrombin activation pathway which is responsible for blood coagulation. In addition, we showed *HSPB9* and *HSPBAP1* to be down-regulated in response to the extreme spring.

Although very few DEGs were detected in the hypothalamus (5 down-regulated), genes encoding Matrix Gla Protein (*MGP*) and Sodium Voltage-Gated Channel Beta Subunit 3 (*SCN3B*) were found to be downregulated during the extreme spring relative to normal conditions. Finally, transcriptomic comparison of heart (12 up-regulated DEGs and 10 down-regulated) revealed genes that are important for the cardiovascular system (e.g., blood coagulation) to be differentially expressed, such as coagulation factor gene (*F3*) and the epidermal surface antigen *FLOT2*. We also found the gene encoding collagen type XIII alpha 1 chain (*COL13A1*) to be up-regulated.

### The snowstorm-induced stress response during incubation
In contrast to the extreme spring, the snowstorm in 2016 was extremely acute (Condition 4 v 3, as described in Methods) causing a series of severe energy challenges and stress responses, which could be explained by birds moving into the ELHS as a coping strategy. In total, we detected 49 DEGs in the hypothalamus (29 up- and 20 down-regulated), 99 in the gonad (35 up- and 64 down-regulated), 1631 in the liver (807 up- and 824 down-regulated), and 315 in heart (140 up- and 175 down-regulated) (Fig. 3C). Genes involved in the circadian rhythm and biological periodicity (e.g., *PER2* and *PER3*) were perturbed across tissues, with *CRY1* additionally down-regulated in the hypothalamus. One of the most consistently up-regulated genes across the hypothalamus, heart and liver (but not testis) was *FKBP5*. Additional stress-related genes were differentially expressed in the heart and liver, e.g., genes encoding Angiotensinogen (*AGT*), Galectin 2 (*LGALS2*), Adrenoceptor Beta 2 (*ADRB2*), Catechol-O-Methyltransferase (*COMT*), along with Haem-Binding Protein 2 (*HEBP2* or *SOUL*) in the hypothalamus. The gene *UCN3*, which encodes urocortin 3 and belongs to the corticotropin releasing factor family, was also down-regulated in the hypothalamus. The gene encoding AGT, involved in the renin-angiotensin-aldosterone system, was shown to be down-regulated in liver[26,27]. The increased expression of *ADRB2* in the liver, is functionally important in the adrenergic system and in metabolism[26] and the reduced expression of *COMT* in heart suggests the moderation of catecholamine degradation such as that of epinephrine[28,29]. In the heart, heat shock proteins (HSP), including *HSPB9* and Heat Shock Protein Family A (Hsp70) Member 5 (*HSPA5*), were up- and down-regulated, respectively, which further suggests their important function during cellular and molecular stress.

Liver DEGs were found to be involved in various metabolic processes, e.g., the regulation of gluconeogenesis and included genes such as Phosphoenolpyruvate Carboxykinase 1 (*PCK1*) and Malic Enzyme 1 (*ME1*) that both produce key enzymes in cytosolic pyruvate metabolism. These differentially expressed genes were found to be enriched for terms including many metabolic functions, such as 'lipid metabolism' and 'carbohydrate metabolism'. Interestingly, significant pathways, such as 'activation of gene expression by SREBF (SREBFP)' and 'TR/RXR activation', are shown to exhibit a positive activation pattern. Taken together, this suggests the overall activation of metabolic gene expression regulation is of endocrinological relevance (Fig. S7C). Significant enrichment of genes with binding sites for transcription factors including HNF1/4, E2F, HIF1, AP4 and MEF2 was also noted (Fig.S7D). Functions of these regulators include determination of cell fate, cell cycle, cell development, energy homoeostasis and mediation of the stress response.

### Hypothalamic-Pituitary-Adrenal axis tissues
In the snowstorm scenario, we additionally investigated 3 tissues to achieve an overview of HPA axis expression patterns in response to the acute

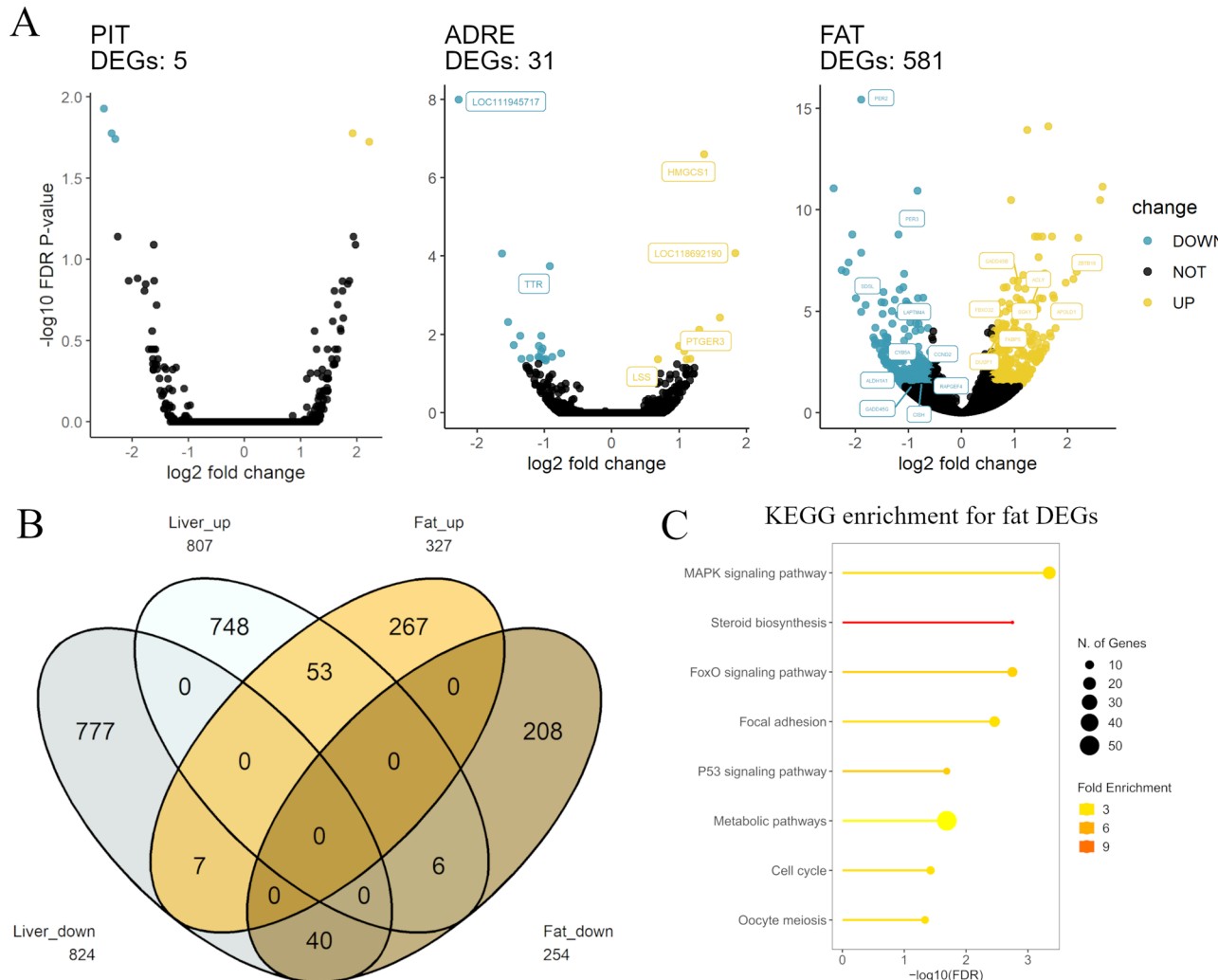

**Fig. 4 | HPA-axis in response to snowstorm. A** DEGs (Differentially Expressed Genes) identified in pituitary, adrenal gland, and fat tissues in response to snowstorm during incubation compared to storm-free controls. **B** Venn diagram showing genes with similar expression patterns in liver and fat. **C** KEGG pathway enrichment for fat DEGs that show either up- or down-regulated pattern in the snowstorm condition.

snowstorm (Fig. 4A). Similar to DEG discovery in hypothalamus, we did not find an extensive number of DEGs in the pituitary (*n* = 5) or adrenal glands (*n* = 31). However, a few genes of importance in stress regulation showed prominent changes, e.g., *PTGER3* and *LSS*. Regarding subcutaneous fat, we found 327 up- and 254 down-regulated DEGs, among which, many are important for energy metabolism. During the snowstorm, 106 DEGs were shared between liver and fat, among which 93 exhibited the same direction of change (Fig. 4B). This interesting pattern further suggests a cohesive response in the metabolic system to cope with the energy needs in different tissues. Enriched KEGG pathways included MAPK signalling pathway, steroid biosynthesis and metabolic pathways (Fig. 4C).

**Comparison of differential gene expression**
The Venn diagrams in Fig. 5 show the unique and overlapping genes between comparisons and tissues. In general, there are not many genes shared between tissues in the same comparison, yet it is to be noted that genes such as *FKBP5* (stress) and *PCK1* (energy demands) suggest a consistent molecular response occurring in corresponding tissues. We also note that differential expression of periodicity-related genes (e.g., *PER2* and *PER3*) is shared across tissues in the storm comparison (Fig. S8). Gene Ontology results using DEGs related to incubation in the storm condition show significantly enriched terms for 'steroid hormones', 'glycolysis/gluconeogenesis' and 'ketone body biosynthetic process'. This sheds light on the switch of physiological preference for energy usage under extreme

snowstorm, whereas DEGs associated with extreme spring show enriched GO terms for acrosome reaction and response to testosterone.

Although the two stress events showed unique gene expression responses, the overlap between the two inclement conditions highlights 90 out of 92 DEGs in the liver with a similar up- or down-regulated pattern (Fig. S9). The two inclement comparisons shared the most significant up-regulated gene, *ME1*. This gene encodes the malic enzyme that catalyses the conversion of malate to pyruvate while generating NADPH in the cytosol, which links the glycolytic and citric acid cycles. ME1 has been reported to rescue human mitochondrial disease complex I mutant cells under nutrient stress conditions[30]. These DEGs are enriched for KEGG terms including 'Pyruvate metabolism' ($P = 8.4 \times 10^{-6}$), 'Glycolysis/Gluconeogenesis', and 'TCA cycle' (Fig. S10), which highlights similar gene expression changes in metabolic pathways.

**Lapland Longspur FKBP5**
One of the highlighted genes from our study, and a gene known to be an important stress-regulator, is *FKBP5*. *FKBP5* is an evolutionary-conserved gene across many vertebrate species and so we decided to investigate the gene in Lapland longspur in a bit more detail. Multiple sequence alignment with various other species shows that the predicted Lapland longspur protein sequence is highly homologous to that of chicken (*G. gallus*) with 93.32% identity (Fig. 6A). The 3-dimensional structure of Lapland longspur FKBP5 protein was predicted by the AlphaFold3 modelling server[31] and is

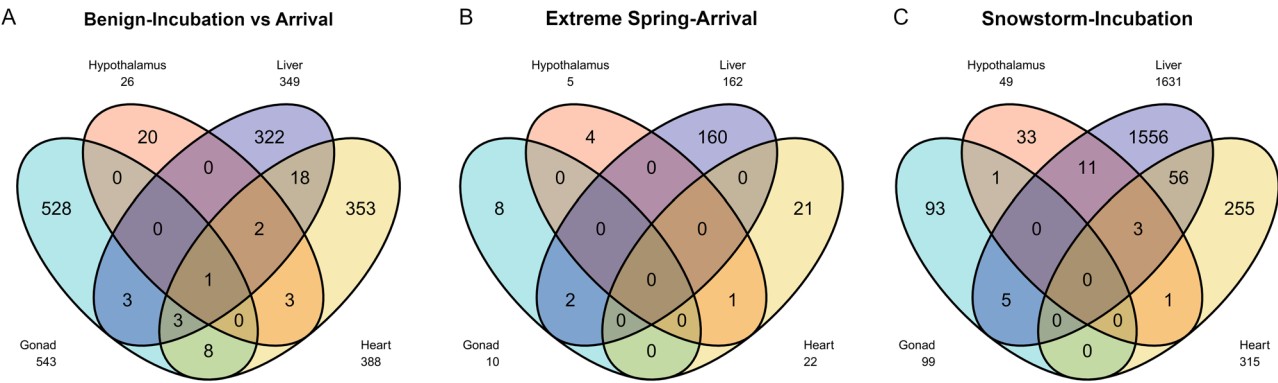

**Fig. 5 | Comparison of differentially expression genes.** Venn diagrams showing the overlap of differentially expressed genes (both up- or down-regulated) detected in different tissues under each stress condition or life-cycle period, including **A** between benign incubation and arrival, **B** extreme spring at arrival, and **C** snowstorm during incubation.

shown in Fig. 6B. A gene interaction network of the human ortholog was also reconstructed and is shown in Fig. 6C. The overall expression dynamics of FKBP5 across tissues and conditions suggest that the *FKBP5* is responsive in different stress scenarios and detectable in various tissue types (Fig. 6D). In addition, we performed network analyses to find the association between gene modules and phenotypes (Fig. S11). Results of weighted gene co-expression network analysis (WGCNA) show that baseline corticosterone concentration is correlated with expression of *FKBP5* ($P = 5.96 \times 10^{-4}$) in the cross-tissue analysis.

## Discussion

In order to understand the effect of extreme weather conditions and how wild, free-living animals respond to them, we generated a genome assembly and performed RNA-sequencing on key tissues in the Arctic specialist, the Lapland longspur, collected during two distinct types of extreme weather conditions (i.e., extreme spring and snowstorm). The high-quality genome assembly enabled an accurate annotation of genomic features and reliable quantification of gene expression changes.

We first investigated gene expression changes occurring between arrival and incubation under normal seasonal conditions, using this as a baseline to understand the transcriptomic dynamics in response to environmental adversity. The major transition between the two life history stages is the physiological adjustment to breeding status, which includes reproductive activity and cardiovascular function. Among other DEGs, *ZP3* shows the most significant down-regulation pattern in testes. *ZP3* encodes a protein essential for the sperm acrosome reaction, especially for binding of sperm to the zona pellucida. ZP3 knockout mice and human genetic mutation in this gene (OMIM:617712) have been shown to exhibit female infertility due to zona pellucida defect[32–35]. Recently, significant *ZP3* expression was found in normal human and mouse testes (spermatogonia and spermatocytes)[18]. A study in a non-mammalian vertebrate, the gilthead seabream (*Sparus aurata*) also demonstrated that the orthologous *ZP3* gene is up-regulated in ejaculated sperm compared to haploid germ cells[23]. Evidence is accumulating for its important role in the male reproductive system, however, the molecular function of a down-regulated *ZP3* in testes is unknown. Another interesting finding was in the cardiovascular system. DEGs in heart exhibit strong enrichment in cardiovascular development, circulatory system development, and positive regulation of vascular associated smooth muscle cell migration (e.g., *IGFBP5*). In addition, the glucocorticoid responsive gene *FKBP5* also expresses at a lower level during incubation relative to arrival in hypothalamus and heart. This supports the hypothesis that during parental stages, the HPA axis activity is attenuated in wild breeding birds to prevent nest abandonment and favour parental care[36]. These DEGs and functional pathways identified under normal conditions provide a foundation to understand the interplay between life history stages and inclement weather events.

The two extreme weather conditions in this study enabled us to compare transcriptomic changes between the two stress scenarios. In liver,

genes with similar up- or down-regulated expression patterns reveal an energy expenditure that is consistent in the two scenarios. For instance, shared up-regulated genes (e.g., *DLAT*, *PDHA2* and *ELOVL5*) are associated with terms such as 'coenzyme biosynthetic process' and 'thioester biosynthetic process'. In liver, one of the most important thioesters is acetyl-CoA, which participates in the tricarboxylic acid cycle to produce energy in the form of ATP or GTP upon oxidisation. Usually, acetyl-CoA is produced via glycolysis of glucose and degradation of fatty acid or amino acid. Genes that positively regulate cellular response to insulin such as *SRC* and *NR1H4* were down-regulated in both inclement conditions, despite the fact that we did not identify statistically significant GO terms using shared down-regulated liver DEGs. Importantly, the large number of differentially expressed genes detected in liver, many of which are important and rate-limiting enzymes in processes of glycolysis and gluconeogenesis (e.g., *PCK1*), were found to associate with related GO terms, which indicates that the dynamic and metabolic activity in liver tissues is providing fuel to cope with the stress-imposed negative energy imbalance.

We have highlighted genes in hypothalamus whose expression is crucially regulated under the inclement conditions studied. *FKBP5* was the most up-regulated gene in hypothalamus during the snowstorm incubation period. This gene is known to have significant relevance as a stress regulator[37,38]. The acute snowstorm immediately affected the function of the HPA axis of Lapland longspur, with an increase in expression of *FKBP5* in the brain. *FKBP5* is a co-chaperone associated with Hsp90, negatively regulating glucocorticoid signalling, which is recognised as a mediator of stress-response. Both nucleotide and protein sequence of *FKBP5* are also highly conserved across species, and a homozygous knockout of FKBP5 in mice was shown to cause abnormal depression or anxiety-related behaviours. Decreased circulating corticosterone levels are also seen in stressed mice[32]. Normally, in response to stress, *FKBP5* expression is increased by elevated glucocorticoids. This correlation is especially interesting in hypothalamus, as it subsequently modulates GR activity by reducing ligand-binding sensitivity and delays the translocation of GR. In turn, GR regulates the transcription of many nuclear expressed genes, including *FKBP5* which is also shown to be determined by the epigenetic context and polymorphism of the locus[38]. High *FKBP5* expression additionally leads to GR resistance and subsequently influences many biological functions and gene transcription, for instance reduced expression of *PCK1*, as we observed[38]. Induced GR resistance can result in an attenuated HPA negative feedback mechanism, whilst a failure to restore the baseline GC level has maladaptive and pathological consequences[39,40]. Collectively, we demonstrate that during the snowstorm, the acute stress was extremely intense, reflected by the activity of HPA axis genes and the significantly induced *FKBP5* expression level in the hypothalamus, eventually leading to a negative balance that had destructive and fatal outcomes. In addition, FKBP5 has been reported to contribute to stress-related cardiovascular risk and metabolic disorders, for instance coronary artery disease and hepatic

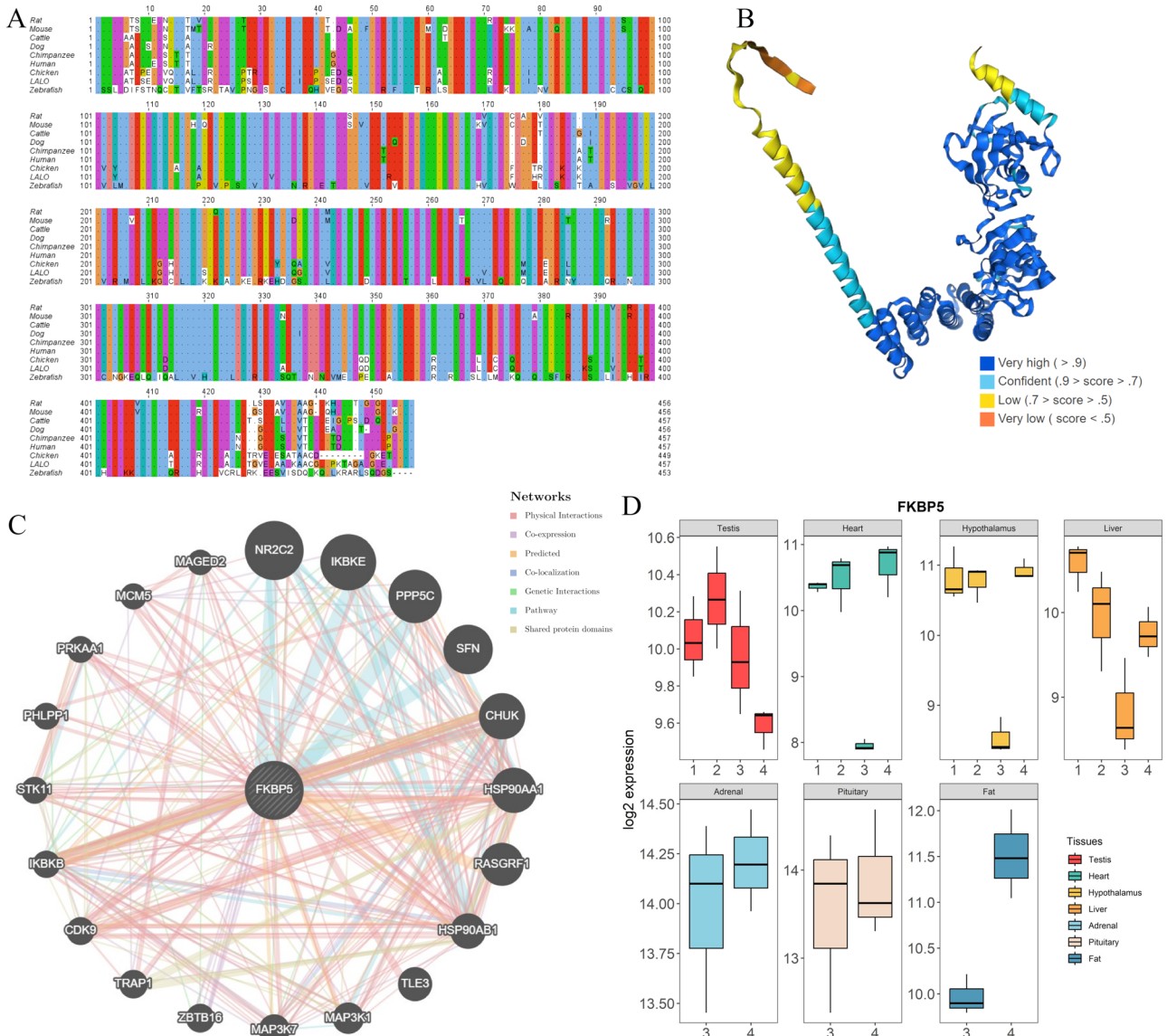

**Fig. 6 | *FKBP5* gene of Lapland longspur. A** Protein sequence alignment of *FKBP5* gene between the predicted Lapland longspur protein (denoted by LALO) and other representative vertebrate species. **B** The predicted 3D model of Lapland longspur FKBP5 protein. Colour scheme represents the confidence of the model. **C** The gene network predicted for human *FKBP5*, with different types of interaction shown by coloured links between genes. **D** Lapland longspur *FKBP5* expression level (log2 transformed) across tissues and conditions. The x-axis shows the 4 studied conditions (see Methods for details), adrenal, pituitary and fat were only examined in the focused snowstorm comparison. The upper and lower quartiles denote the border of box plots, the bar inside the box shows the median value, the vertical lines display the minimum and maximum of the expression level.

disease[38,41,42]. Expressional changes of *FKBP5* across different tissues suggest its vital molecular functions in modulating several stress-related biological pathways.

Lapland longspurs had to fight against the chronic and prolonged inclemency of weather in the extreme spring scenario by triggering the ELHS, and actively trading off their breeding windows for the chance of survival. The reduced expression level of *ZP3* and *FMN2* in testes sheds light on the gene expression dynamics occurring during reproductive curtailment. In our analysis between benign life history stages, *ZP3* was the most significantly down-regulated gene in testes of incubating birds. Recent evidence of up-regulated *ZP3* expression was found in testes of some vertebrates[18,23]. Similar high expression was found for *FMN2* in early spermatids in a recent human single-cell study[33]. *FMN2* encodes a protein crucial for actin binding and cell polarity (mostly associated with female infertility)[34,35], as well as being an important component involved in stress-induced cell-cycle arrest in human cells[43]. In addition, expression of

*TBC1D8* is increased in testes, with recent studies suggesting its function in cell death as a new cell apoptosis inducer[44], which may mark it as a potential key protein during curtailment of reproduction. Our discovery of low expression of *ZP3* (and *FMN2*) in reproductively curtailed passerine birds further demonstrates that reduced gene expression can cause arrested spermatogenic maturation which underlies the impaired male reproduction in extreme spring. We therefore hypothesize that the testicular expression of *ZP3* is crucial for the initiation of spermatogenesis, with function rapidly reducing upon entering the incubation period. To our knowledge, this study shows, for the first time in wild living *aves*, expression of *ZP3* has fundamental function during spermatogenesis, and reduced expression may indicate reproductive inhibition. Compared to previous studies on the hypothalamic-pituitary-gonadal response to stress[45,46], novel genes identified in our study complement the few discovered genes in the male transcriptomic stress response and expand our understanding of the influence of a prolonged stressor on body systems.

We noted that the expression of *FKBP5* was not significantly regulated in any tissue during the extreme spring, which may suggest that the HPA axis had likely acquired a temporal balance with improved plasticity and had adapted to perturbations at the sampling time point. These temporal dynamics could be achieved by many possible mechanisms (e.g., corticosterone binding globulin)[11,47]. The fitness of the birds during the extreme spring may also be explained by the previously-studied association between HPA flexibility and *FKBP5* expression[48]. This pattern agrees with the "leave-it" strategy as previously proposed[5,8]. Although FKBP5 levels can be a proxy for a chronic stressor[47], our study highlights the importance of considering the interplay of life history stage, extreme environmental conditions, coping strategy, and temporal gene expression fluctuations. It is important to note that during the life history cycles the gene-environment interaction is complex and dynamic. Other factors such as carryover effects of previous states[49], seasonal interactions[50], confounding effect of the year, and other environmental or social trade-offs can altogether contribute to fitness outcomes, although their effect may be smaller compared to the specific extreme weather events[49]. The two extreme weather scenarios in this study offer unique insights, but may not fully represent the impact of other types of extreme weather events. Nonetheless, we believe the molecular investigation of these specific events provides valuable context for future studies on more generalised situations.

When comparing the two scenarios from an allostatic load perspective, they act similarly as a source of allostatic load with limited food availability and higher thermoregulation demand, which limits the amount of external energy availability in the environment (EG) and increases the cost of maintaining minimum day-to-day homoeostasis (EE) as well as energy required to make predictable changes under ideal conditions (EI), which triggered significant negative energy balance[9]. Spring is thought to provide higher energy supplies in food compared to winter, however, the chronic effect of an extremely cold period imposes constant challenge on the energy balance, on top of the expenditure of life history stage transition (e.g., migration and pre-breeding). The ELHS was thus adopted by Lapland longspurs, which in turn suppressed the regular LHS (i.e., breeding). Through these series of physiological and behavioural stress challenges, the birds successfully managed to reduce their negative energy balance and reach a favourable outcome of survival. However, this was at the cost of a failed breeding attempt. In comparison, the extreme snowstorm is expected to have caused immediate overloads by dramatic energy expenditure via the environmental perturbation (EO) that is often expected during adversity along with glucocorticoid secretion. This negative energy balance resulted in an acute effect of EE+EI+EO>EG, with individuals failing to reduce the allostatic load eventually suffering pathological effects and, to a greater degree, death. This additionally supports the double-edged sword theory of glucocorticoid function and other allostasis mediators, which play protective and adaptive roles in the stress response, yet can also have damaging effects.

Beyond the above-mentioned genes, further genes are of interest, including those which contribute to the comprehensive stress response that induces physiological, behavioural and phenological changes. For instance, genes (e.g., *FKBP5*) that have been reported to be stress-related in human studies are associated with the adrenergic system, bipolar condition and the renin-angiotensin-aldosterone system (RAAS). These genes were detected mostly in heart (*COMT* and *LGALS2*) and liver (*AGT*, *ADRB2*, and *LGALS2*) in our comparisons[26]. We showed that heat shock protein (HSP) genes, along with associated genes, to be differentially expressed across different conditions and tissues. Of special interest, *HSPB9* is down-regulated in the heart tissue during the benign breeding cycle comparison and up-regulated during the snowstorm. This gene also showed reduced expression in the liver after experiencing extreme spring conditions. In addition to the protective function of the HSPB family when exposed to stress conditions, HSPB9 is reported to play a role in cytoskeletal stability under stress[51,52]. Gene *HSPA5* is lowly expressed in response to snowstorm in heart. This protein is known to be involved in endoplasmic reticulum (ER) homoeostasis, especially in the unfolded protein response (UPR).

When under accumulated or prolonged ER stress with protein load, this molecular pathway can lead to ER-associated degradation or apoptosis[53]. A previous study suggested that in migratory passerine birds, up-regulated *HSPA5* may contribute to the response to migration-induced stress[54]. In addition, we also found *ZBTB16* to be upregulated in liver and heart in response to the snowstorm. Gene *ZBTB16* encodes a zinc finger transcription factor that responds to corticosteroid regulation, and shows a robust induction by GR activation or stress exposure in human, mouse, and bird studies[46,55,56].

In summary, here we present a high-quality genome assembly of the Lapland longspur (*Calcarius lapponicus*), which has enabled a comprehensive transcriptomic study to investigate gene expression changes in key tissues, occurring in response to environment stress caused by extreme weather events at different stages of the Lapland longspur breeding cycle. Biological processes such as activation of HPA axis, reproductive function and energy regulation are shown to be central to the adaptive changes required in the host, with genes including *FKBP5*, *ZP3* and *ME1* being key players in these responses. The two specific scenarios showcased the transcriptomic investigation of stress response in wild-living birds, providing valuable context for future studies on more generalised situations. This study provides data fundamental to our understanding of how species will have to be able to adapt to the rapidly changing climate we all now face.

## Methods

### Lapland longspur genome assembly

A pectoralis muscle sample from a female Lapland longspur (*Calcarius lapponicus*) was selected for generating the genome assembly. In order to generate a high-quality genome, high molecular weight DNA (50 to 100 kb) was extracted which was subsequently used for long-read PacBio library preparation and sequencing. Sequencing of the genome was performed with PacBio Sequel II SMRT Cell, and we subsequently assembled the sequences into scaffolds by using Wtdbg2[57]. In addition, Omni-C scaffolding was carried out by Dovetail Genomics (CA, USA). The Omni-C protocol provides a sequence-independent approach: chromatin was fixed in place and then optimised DNase I was employed to digest the chromatin. The DNA was sequenced on an Illumina HiSeqX platform to produce $2 \times 150$ bp paired-end libraries, yielding a coverage of around 30X. The HiRise pipeline was then used to identify sequence joins and produce scaffolds[58]. The species in this study is not a model species nor from an inbred line, therefore the heterozygous nature of the genome could potentially influence the analyses, thus we employed Purge_Dups (1.2.5) to detect and remove false duplications such as heterotype sequence with default features[59]. We performed gene annotation using the information from the RNA-seq data to refine splice junctions and Iso-seq data from a closely-related species (*Z. leucophrys*) to provide cross-species evidence by using GMAP (>90% identity)[60]. The clean sequence was mapped to the genome, with the cross-species evidence used as a guide reference. Predicted splice junctions were then quality-controlled by Portcullis (v1.2.0) (https://github.com/EI-CoreBioinformatics/portcullis). Multi-sample transcripts were combined to robustly predict transcripts by majority voting with PsiCLASS (v1.0.2)[61]. Coding probability was estimated by CPC2 (version 0.1)[62] using the longest transcript with '-r' function to check the reverse strand. In total, we estimated 17,096 coding genes. The gene features were subsequently used for mapping and quantifying the RNA-seq reads.

### Evaluation of genome assembly

In order to assess the quality of the Lapland longspur genome, basic quality statistics of the assembly were computed by measuring the total size, count of scaffolds/contigs, contiguity (such as N50/N90, or contig N50/N90), and GC content. Repeat elements like transposable elements were trained and identified by RepeatMasker (v4.1.2) and Repeatmodeler (v2.0.2)[63,64]. We quantified the completeness of expected conserved genes in the genome using Benchmarking Universal Single-Copy Orthologs (BUSCO), with

databases of vertebrata_odb10, aves_odb10, and passeriformes_odb10[65]. Mitochondrial genome (MT) sequence was detected as the MT scaffold. MitoZ software (version 2.2)[66] was used to detect the circular MT sequence and annotate the genes. We reordered the sequence to anchor the proximal starting position of the circular sequence by using the MT genome of a closely-related species, the Red Crossbill (*Loxia curvirostra*, NC_025623.1)[16], as well as using the chicken MT sequence (GRCg6a, NC_040902.1) and Japanese Grosbeak (*Eophona personata*, KX812499)[15] MT genome for verification.

## Comparative genomic analyses
As for comparative analyses between the genome of Lapland longspur and that of other avian species, the Mummer aligner (v3.2.3)[67] was used to align our genome to the reference genome of chicken (*Gallus gallus*, GRCg6a) and zebra finch (*Taeniopygia* guttata, bTaeGut1.4.pri, GCF_003957565.2). Due to lack of karyotype information for the Lapland longspur genome and the mapping uncertainty for micro-chromosomes in birds, we performed the comparative analyses and visualised the macro-, micro- and sex chromosome results separately, shown by scaffolds 1 to 23 in Fig. 2A and the other major scaffolds in Fig. S1C, D. The chromosome assignment of our assembly (Supplementary Table S3) was based on zebra finch and chicken chromosomes. The majority of scaffold 4 was assigned as the Z chromosome. Scaffolds 5 and 18 were assigned as chromosomes 1A and 4A based on the zebra finch karyotype. Selected micro-chromosomes of *T. guttata* (chromosome 23–29) were aligned separately and shown to map to Lapland longspur scaffolds 11, 12, and 195. The circular visualisation of the scaffolds was conducted using the R package circlize (0.4.15)[68], GC content and repeats were summarised in 200 kb windows, and Ns or gaps were summarised in 50 kb bins. A DOT plot was used to visualise the cross-species alignment by adapting R code from dotPlotly (https://github.com/tpoorten/dotPlotly).

## Sample collection
Tissues from 12 male Lapland longspurs were collected across the breeding season in 2013 and 2016. Samples were collected from four breeding sites on the north slope of Alaska, USA. Birds were captured with seed-baited potter traps or mist nets. Blood samples were collected from the alar vein with a 26-gauge needle and surface blood was collected by heparinized microcapillary tube within 3 minutes of capture. The birds were sedated with isoflurane and euthanized by rapid decapitation (3 min 20 s ± 52 s post capture). After euthanasia, tissues were dissected, wrapped in aluminium foil, frozen on dry ice, placed into labelled plastic bags, and kept frozen on dry ice until they were stored in a −80 °C freezer upon returning to the laboratory. Samples were later shipped on dry ice to the Roslin Institute, University of Edinburgh, UK where they were stored at −80 °C until use. Blood samples were centrifuged at 13,000 rpm for 5 min, the plasma was aspirated using a Hamilton syringe, transferred to a microcentrifuge tube, and frozen at −30 °C till assay. Corticosterone concentrations were measured by radioimmunoassay as described by Wingfield et al. (1992) and Krause et al. (2015)[69,70]. The fat score and body weight (g) were recorded at capture. Fat score of furcular and abdominal fat was measured on a scale from 0 (lean) to 5 (fat) and combined as previously described[7,71].

## RNA preparation and sequencing
Four environmental conditions at two life-history stages (arrival and incubation) were investigated: (1) 2013 arrival - extreme spring, (2) 2016 arrival - spring and storm-free, (3) 2016 incubation - storm-free, (4) 2016 incubation - snowstorm. For each condition and tissue, we sequenced three replicates to ensure statistical power. RNA samples from the testis, heart, hypothalamus, and liver were prepared by homogenisation in TRIzol reagent (Invitrogen, Paisley, UK) and extracted following the protocol based on the Direct-zol RNA Miniprep kit (Zymo Research USA), with a final elution of RNA in RNAse-free water. Following RNA extraction, total concentration of RNA was determined via the nanodrop

spectrophotometer (Thermo Fisher, USA). After library construction and PolyA selection, RNA was sequenced on an Illumina NovaSeq platform, producing $2 \times 150$ bp paired-end reads. Forty-eight samples were sequenced in total.

Additionally, in order to validate the effect of snowstorm along the HPA axis and in HPA target tissues, we conducted additional RNA-seq on pituitary gland, adrenal gland, as well as subcutaneous fat tissues. Here we focused on the comparison between environmental conditions (3) and (4); birds captured during the incubation period during a snowstorm relative to normal conditions. The RNA was prepared and sequenced using the same methods mentioned above. An additional 18 samples were thus sequenced.

## Detection of differentially expressed genes
The raw RNA-seq reads were quality checked using FastQC with default parameters[72], and adaptors were trimmed using Trimmomatic (0.39)[73]. Clean reads were mapped to the Lapland longspur genome using STAR (v2.7.8a_2021-03-08)[74] with the following parameters: --outFilterType BySJout --outSAMunmapped None --outSAMtype BAM SortedBy-Coordinate --runThreadN. Quantification of gene expression level was performed by assigning the reads to genomic features using featureCounts[75], which is part of the Subread package. We counted reads meeting the following criteria, both ends of paired reads aligned (-B), paired-end distance thresholds of 5000 (-P -D 5000), count fragments of reads (-p), and feature type to count is 'exon'. Expression count matrix was then processed by the R packages DESeq2[76] and edgeR[77]. We pre-processed the data for sample normalisation and to remove the poorly aligned genes, where genes with an average read count of at least 1 per sample were included. Differentially expressed genes (DEGs) with considered with $P < 0.05$ and fold change >1.5. DEGs related to stress response were identified by comparing the samples from inclement to normal weather conditions for each tissue. Arrival during extreme and normal spring (Condition 1 v 2) was compared along with samples from birds incubating during storm and normal conditions (Condition 4 v 3). DEGs associated with the annual life-cycle were detected by comparing birds during the incubation period with those collected during arrival at the breeding ground - both collected during storm-free, normal spring (Condition 3 v 2).

## Weighted gene co-expression network analysis
In order to find clusters of highly correlated genes in our dataset and their association with measured traits, we performed weighted gene co-expression network analysis (WGCNA) (version 1.71)[76]. First, the complete dataset of 48 samples was analysed and a network was constructed with a soft-thresholding power of 11 for signed network. Subsequently, acknowledging the biological heterogeneity between tissues, we performed an additional module analysis across the four tissues. Pair-wise Pearson correlation was calculated between the identified modules and the selected traits, for which a student asymptotic P-value was also calculated.

## Functional annotation
The differentially expressed genes were compared between tissues and their function further investigated. Venn diagrams were generated for each type of environmental stress to find unique and overlapping DEGs across the four tissues using the "ggvenn", "nVennR" and "euler" packages in R. To understand the biological importance of genes of interest, we used either chicken or human homologous genes for functional analyses due to availability of better gene functional annotation. The differentially expressed gene sets were analysed for Gene Ontology (GO) enrichment and KEGG pathways annotation based on chicken orthologous genes using shinygo 0.77 (http://bioinformatics.sdstate.edu/go/). The FDR *P*-value and Fold Enrichment were calculated to measure the overrepresentation of genes. Homologous human genes were analysed by Ingenuity Pathway Analysis (IPA) software (QIAGEN Inc., https://digitalinsights.qiagen.com/IPA)[78] to reveal biological pathways and functions pertaining to the DEGs identified in our analyses. The P-value was calculated using the right-tailed Fisher Exact Test with Benjamini-Hochberg correction (threshold $P < 0.05$).

Enriched transcription factor binding sites were identified using WebGestalt (WEB-based GEne SeT AnaLysis Toolkit) algorithms (http://www.webgestalt.org/)[79]. Over-representation analysis (ORA) was carried out using the network functional database to identify transcription factor targets. The 'chicken genome' was used as the background reference. Results showing an FDR $P < 0.05$ were considered significant.

## FKBP5 gene network and protein structure prediction

As a specific differentially expressed gene of interest, a gene network was generated for FKBP5 using the association network prediction server GeneMANIA (3.6.0)[80]. Based on the known function of human FKBP5, the programme generated a functional association gene network by a label propagation algorithm, which shows how the genes interact with each other. A multi-species alignment was performed using 8 other vertebrate FKBP5 protein sequences retrieved from NCBI including *Homo sapiens* (NP_004108.1), *Pan troglodytes* (XP_001172420.1), *Canis lupus familiaris* (XP_538880.2), *Bos Taurus* (NP_001179791.1), *Mus musculus* (NP_034350.1), *Rattus norvegicus* (NP_001012174.1), *Gallus gallus* (NP_001005431.1) and *Danio rerio* (NP_998314.1), multiple sequence alignment was performed using the Clustal Omega aligner[81] with default settings. The results were visualised by Jalview (version: 2.11.2.4)[82]. A neighbour-joining (NJ) tree was also generated by the build-in function for the purpose of comparing pair-wise sequence identity and detecting the most closely related protein sequence. The 3D modelling of FKBP5 protein was conducted on the AlphaFold3 online server (https://www.alphafoldserver.com, accessed May 10th, 2024), using the default protein modelling[31].

## Statistics and reproducibility

Statistical analyses were conducted to identify differentially expressed genes under the two extreme weather conditions and life history stages. DEGs were identified using a significance threshold of FDR $P < 0.05$ and Fold Change > 1.5. Tissue-specific DEGs were determined for each comparison. Weighted Gene Co-expression Network Analysis (WGCNA) identified gene modules correlated with traits, using pair-wise Pearson correlation with asymptotic P-values and correlation coefficients. KEGG pathway enrichment analyses were performed with an FDR $P < 0.05$ threshold. A total of 66 RNA-seq datasets were generated across four primary tissues (testis, heart, hypothalamus, and liver) and three additional tissues (pituitary, adrenal glands, and fat). Each condition and tissue included three biological replicates, individually sampled from distinct weather conditions and life-history stages. All datasets and codes are publicly available for reproducibility.

## Reporting summary

Further information on research design is available in the Nature Portfolio Reporting Summary linked to this article.

## Data availability

Sequence data presented in this paper have been submitted to the NCBI public database under the following accession numbers: Calcarius lapponicus genome assembly - JBBFKN01 (GenBank accession: genome assembly - GCA_039654755.1, MT sequence- CM077482.1); RNA-seq data - PRJNA1023066[83,84]; Iso-seq data - SAMN31228560, SAMN31228561, SAMN31228562, under PRJNA889240[85]. Phenotypic measurement data is also accessible through the Zenodo repository (https://doi.org/10.5281/zenodo.14191540)[86].

## Code availability

Additional scripts used to generate results and figures are available in the GitHub repository (https://github.com/wzuhou/LALO_scripts) and Zenodo (https://doi.org/10.5281/zenodo.14183369)[87].

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

## Acknowledgements

This work was partly funded by the Biotechnology and Biological Sciences Research Council (BBSRC) under grant number BB/V001647/1 to J.S. and S.L.M. This work was also supported by the National Science Foundation Office of Polar Programs ARC 0909133 and Integrative Organismal Systems IOS 1558049 to J.C.W. and Roslin Institute Strategic Grant funding from the UK Biotechnology and Biological Sciences Research Council (BB/P013759/1) to S.L.M. J.C.W. would like to acknowledge the University of California, Davis Endowed Chair in Physiology. Additional RNA-sequencing was supported by the Roslin Institute Early Career Grants scheme to Z.W. to fund the project entitled "Discovering genes for resilience: Uncovering the transcriptomic stress response to inclement weather conditions" through the ISPs (Institute Strategic Programmes). The authors would like to acknowledge the services of Edinburgh Genomics (Edinburgh, UK) for carrying out the RNA-Sequencing and Dovetail Genomics (CA, USA) for genome assembly. We would like to thank Helen E. Chmura and Jeffrey Cheah for their help in the field and logistic support from the staff at Toolik Field Station, The University of Alaska Fairbanks, USA. We thank Brian Barnes and Jeannette Moore for logistical support while in Fairbanks, Alaska, USA. For the purpose of open access, the author has applied a Creative Commons Attribution (CC BY) licence to any Author Accepted Manuscript version arising from this submission.

## Author contributions

Conceptualisation: S.L.M., J.S.K., J.C.W., J.S., and Z.W.: Sample collection: S.L.M., J.K., J.H.P., and J.C.W.: Laboratory work: V.R.B., A.M.A.R., J.S.K., K.M., and Z.W.: Genome assembly and RNA-seq analyses: Z.W., M.M.H., and J.S.: Visualisation: Z.W. and J.S.: Supervision: J.S. and S.L.M.: Funding acquisition: J.S., M.M.H., and S.L.M.: Writing—original draft: Z.W. and J.S.: Writing—review & editing: Z.W., J.S., S.L.M., J.C.W., and J.S.K.

## Competing interests

The authors declare no competing interests.

## Ethics approval

The samples used in this study were collected from the male adult Lapland longspur (*Calcarius lapponicus*). The work was approved by the University of California, Davis, USA Institutional Animal Care and Use Committee (AICUC) under protocol 19758, United States Fish and Wildlife Service - Federal MB90026B-0 and The Animal Welfare and Ethical Review Body at the Roslin Institute, The University of Edinburgh, UK. We have complied with all relevant ethical regulations for animal use.
