## [Transparent Peer Review file · Communications Biology]

Response strategies to acute and chronic environmental stress in the arctic breeding Lapland longspur (*Calcarius lapponicus*)

Corresponding Author: Dr Zhou Wu

This manuscript has been previously reviewed at another journal. This document only contains information relating to versions considered at Communications Biology.

Version 0:

Reviewer comments:

Reviewer #1

(Remarks to the Author)

The manuscript is well prepared, with comprehensive methodology and clearly reported results. Indeed, the genome assembly and gene expression profiles provide comprehensive resources for future studies.

I have only a few minor suggestions regarding the discussion and some clarifications needed regarding the methods that I hope they will help the authors to further enhance this ms:

Main

L.89 Here i miss a bit the justification for choosing specific tissues (e.g., testis, heart, hypothalamus, liver) over others. Did you select as many tissues as you could for exploratory runs of gene expression, or is there a specific reason/hypothesis behind the selection? It is not fully elaborated. Providing more detailed rationale for tissue selection and how these choices directly contribute to the study's objectives would strengthen the aims section and the ms overall.

Results

I cannot seem to find information about the base-pair quality of this reference genome? That would be helpful.

L.244 One thing that the authors should think about is if they expect that FKBP5 has the same role in all tissues where is expressed. Surely it is all about stress when expressed in the hypothalamus, but what about the heart and liver? Could it be related to body condition?

Discussion

A potential concern here is that the study may oversimplify gene-environment interactions, that are typically complex and multifactorial. So some significant life-history outcomes are potentially attributed to a few genes without considering other factors. Additionally, the fact that you euthanise birds does not allow to sample the same birds at different life stages (that could lower variation between samples) but even so, keep in mind that carry-over effects from the non-migratory state, or conditions during migration, could have severe effects on the individual level and be involved in gene expression (<https://doi.org/10.1371/journal.pone.0141580>, <https://doi.org/10.1093/gbe/evad061>, <https://doi.org/10.1002/ecy.3938>). Of course, on the other hand, you might be fine as carryover effects may be minimal relative to effects of conditions experienced on breeding grounds, or buffered by them (<https://doi.org/10.1038/s41598-020-80341-x>, <https://doi.org/10.1002/ece3.8588>). I personally think that this is the case (conditions experienced by Lapland longspurs upon arrival at breeding grounds are stronger than any carry-over effects) but such effects are not simple and you should at least bring some caution to that aspect on the discussion.

From a scan of all of the DEGs, I spotted that some heatshock genes were in the lists. This is pretty interesting and such

stress proteins are within the scope of the ms but are totally missing from the discussion. Specifically, HSPB9 was downregulated in the heart tissue between life history stages and upregulated in the snowstorm-induced stress response during incubation. The same gene was also downregulated in the liver when tested for the extreme spring. Another heat-shock gene only expressed in the liver tissue was HSPBAP1, downregulated both between the life history stages and the extreme spring arrival condition. Finally, HSPA5 that was downregulated in the heart during the incubation-snowstorm scenario is also very interesting. HSPA5 is known to regulate ER homeostasis (<https://doi.org/10.1111/brv.12667>) and has its upregulation in lean individuals has been suggested as a response to the physiological challenges that migratory birds face (<https://doi.org/10.1007/s00360-023-01529-x>). Overall, heat shocks seem to play a major role in stress response (<https://doi.org/10.1046/j.1461-0248.2003.00528.x>) and therefore it could be good to add a few statements in the discussion. there are many instances where such a discussion could be accommodated, e.g. L.376 where you almost touch the subject by stating that FKBP5 is a co-chaperone associated with Hsp90. Or in L.419 as another possible stress response mechanism. Or at least in L.445 as further genes of interest.

L.375 ...and is suggested to be central in the regulation of HPA (<https://doi.org/10.1016/j.yhbeh.2021.105038>). Or this reference could work better in L. 424-425.

Methods

While the study draws strong conclusions about the role of specific genes (mainly about FKBP5) in response to stress, it does not sufficiently explore potential relationships between genes. For example, i understand that you chose to construct a gene network for FKBP5 since it consistently popped up as differentially expressed and was the most up-regulated gene in the hypothalamus, but maybe a Weighted correlation network analysis would be more appropriate and provide more insight on the roles of different genes?

L536 blood plasma was also collected and stored till assay. But what assay? Nothing is mentioned about running any blood samples. I suppose that this is not needed here?

L.617 The mitogenome was also assembled, the access of which should also be documented. In the interest of reproducibility, it is advisable to create a GitHub (or similar) page containing the commands and parameters used and the entire pipeline used for genome assembly and RNA-seq/DEG analysis. Some values are mentioned (e.g. L561) but maybe keeping them in a single place will facilitate transparency and accessibility for future researchers.

Fig.2: Maybe a snail plot would be a more efficient way to describe/summarise the assembly (e.g. <https://doi.org/10.12688/wellcomeopenres.15679.1>). This way you can also report BUSCO results and scaffold statistics, so Table 1 can be moved to the supplement making the visualisation of your results more clear. Maybe panel C can be moved to the supplement too.

Reviewer #2

(Remarks to the Author)

The present manuscript by Wu and colleagues aimed first to assemble the Lapland longspur genome, and then to determine the effect of environmental stressor (snowstorm) at different physiological stages (upon arrival and breeding). Several tissues were collected and subjected to RNAseq which allows the identification of many DEGs.

Although the study is very interesting and difficult (capturing wild birds, etc.), several concerns need to be addressed:

- The study is very descriptive.
- Without any phenotypic data (body weight, fat content, liver weight, stress indicator, reproductive measurement), the study is very limited and does not provide much except variation in transcriptomics.
- Some birds were captured in 2013 and other in 2016. Where there any effect of year? Also what was the age of birds? There were 20 birds in total, how many birds per season and physiological status?
- Birds were captured with seed-baited potter traps- this might induce additional stress. How have authors controlled for that?
- Line 173, in what tissue FBXO32 was up regulated?
- Line 249, UCN3 is urocortin 3. It belongs to CRF family, but it not CRF.

Reviewer #3

(Remarks to the Author)

General comments:

This is a nicely conceived and very well-written study that I enjoyed reading. I agree with the authors that these data are in high demand to provide a mechanistic, molecular, framework informing on how birds acclimate to environmental adversity. Some aspects of your work may require further attention, however. I have outlined some general remarks below, which are followed by a short list of line-specific comments that mostly pertain to style. I hope you will find these remarks useful as you prepare a revision.

1. The ms, valuable as it is, comes across as somewhat anecdotal as you draw on two case studies to make very affirmative conclusions about the molecular responses to extreme weather events and energy stress. It may well be that you are correct, but without replication there is really no way for us to gauge the reproducibility and generality of your data. I think you need to formally acknowledge this in the ms.

2. While I do not mind the snowstorm example, I struggle more with the “late spring” case study where you contrast an adverse spring with a normal spring occurring three years later. This goes back to the issue of replication, because your study design inherently confounds year and weather event. It would be useful with additional data demonstrating similarity of gene expression profiles across a range of normal springs.

3. Your ms nicely sets the stage for transcriptional changes used to counter environmental perturbations and highlight two case studies answering to this. However, the seasonal progression result appears somewhat disparate in this regard and I was not convinced that it fits into the overall narrative. It has great value still, but to really push the points you make in the Introduction, I advise you consider packaging the seasonal comparison into the online supplements. I understand that this could be useful for understanding the changes occurring in response to environmental adversity, but with the current packaging this does not really come through that clearly even after having read your discussion on the matter.

Line-specific comments:

L35: To make this point, you may want to defined extreme weather conditions, since many would argue that “yes, such are common, but are they really that common?”.

L35-38: Unclear, please revise.

L49: I agree and disagree. This is absolutely true for a cold challenge, but in the context of heat stress, water may become limiting much earlier than energy ever becomes an issue.

L58-59: I advise you unpack this a bit more to set the stage for your study.

L79: This will sound minor to any non-specialist and so I suggest you provide more biological context of this delay.

L97, L111, L132 (- L146): I would place these three paragraphs in the Methods, because they do not really answer to your research problem.

L346-355: This is largely repetitive of the Introduction and can be removed.

Version 1:

Reviewer comments:

Reviewer #1

(Remarks to the Author)

The authors did a great job revising the manuscript and I believe that results and conclusions are now more clear. I just have two comments both regarding the correction of typos.

L.95 change accesss to assess

L.502 change differently to differentially

Reviewer #3

(Remarks to the Author)

Dear authors,

I have now read the revised version of your ms COMMSBIO-24-4415A on the molecular regulation of the responses to extreme weather events in Lapland longspurs. I was Reviewer 3 on the original paper.

I raised initial concerns about the paper being somewhat anecdotal on account of the inherent lack of replication. I still believe that it is. However, the revisions have seen the text and conclusions becoming much more nuanced, which serves the important purpose of making the merits of this work come through more clearly. Overall, the paper has improved substantially in this process - not the least by the changes made to the many insightful comments by R1 and R2.

While I still maintain some reservations, I do not think that the main purpose of revising any manuscript is to please a reviewer in every regard. To me, the added balancing of insights and limitations is plenty sufficient and I will recommend the paper is accepted.

Response to Reviewers' comments

We would like to thank the reviewers for their time assessing our manuscript, and for their insightful suggestions which have helped improve the paper. Please find our point-by-point responses to specific comments in red font below. We have addressed all the comments in detail, and showed track-changes in the manuscript.

Reviewer #1 (Remarks to the Author):

The manuscript is well prepared, with comprehensive methodology and clearly reported results. Indeed, the genome assembly and gene expression profiles provide comprehensive resources for future studies.

I have only a few minor suggestions regarding the discussion and some clarifications needed regarding the methods that I hope they will help the authors to further enhance this ms:

We really appreciate the insightful comments and suggestions from the reviewer. We address all the comments in detail, and show track-changes in the manuscript. Please see our response to specific comments in the following sections (shown in red font).

Main

L.89 Here i miss a bit the justification for choosing specific tissues (e.g., testis, heart, hypothalamus, liver) over others. Did you select as many tissues as you could for exploratory runs of gene expression, or is there a specific reason/hypothesis behind the selection? It is not fully elaborated. Providing more detailed rationale for tissue selection and how these choices directly contribute to the study's objectives would strengthen the aims section and the ms overall.

R1: We collected different tissues based on previous studies of the physiological changes in response to extreme weather events in the Arctic. For instance, we include the hypothalamus, pituitary and adrenal gland as these are components of the stress axis. As reproduction is delayed on arrival at breeding grounds in the year when there were extreme weather conditions, we collected the testes. We have previously established that hematocrit levels and body condition are affected during extreme spring, thus we included heart, liver and fat tissues. We have now included the justification in Line 95-97 (manuscript references: 5,7,11).

Results

I cannot seem to find information about the base-pair quality of this reference genome? That would be helpful.

R2: A snail plot with the assessment of the genome quality has now been generated - please see Figure S2. In addition, we used short-read sequencing of a different individual, and managed to reach 98.96% mapping rate, with a mean mapping quality of 43.81 and a general error rate of 0.0191. This information is now included in Line 118-121.

L.244 One thing that the authors should think about is if they expect that FKBP5 has the same role in all tissues where is expressed. Surely it is all about stress when expressed in the hypothalamus, but what about the heart and liver? Could it be related to body condition?

R3: We agree that FKBP5 expression in different tissues can have different molecular functions that are stress-related. For instance, in the heart, FKBP5 has been reported in humans to be associated with cardiovascular risk, such as coronary artery disease; similarly, FKBP5 has been shown to contribute to metabolic disorders and liver disease. Expressional changes of FKBP5 in multiple tissues suggest its

vital roles in modulating several molecular pathways related to stress. This part of the discussion is now included in Line 427-431.

Discussion

A potential concern here is that the study may oversimplify gene-environment interactions, that are typically complex and multifactorial. So some significant life-history outcomes are potentially attributed to a few genes without considering other factors. Additionally, the fact that you euthanise birds does not allow to sample the same birds at different life stages (that could lower variation between samples) but even so, keep in mind that carry-over effects from the non-migratory state, or conditions during migration, could have severe effects on the individual level and be involved in gene expression (<https://doi.org/10.1371/journal.pone.0141580>, <https://doi.org/10.1093/gbe/evad061>, <https://doi.org/10.1002/ecy.3938>). Of course, on the other hand, you might be fine as carryover effects may be minimal relative to effects of conditions experienced on breeding grounds, or buffered by them (<https://doi.org/10.1038/s41598-020-80341-x>, <https://doi.org/10.1002/ece3.8588>). I personally think that this is the case (conditions experienced by Lapland longspurs upon arrival at breeding grounds are stronger than any carry-over effects) but such effects are not simple and you should at least bring some caution to that aspect on the discussion.

R4: We fully acknowledged the concern raised, and we do agree that we are only focusing on single timepoints and the immediate effect caused by them. It would be interesting to track the longer effect on the same group of birds, however this is challenging given the sampling approach required as these are wild free-living birds (for instance, to sacrifice the birds for RNA-seq tissues), and the interaction between these events and future adversity. Factors such as carryover effects of previous states (e.g., during migration), seasonal interactions, confounding effect of the year, and other environmental or social trade-offs can no doubt collectively contribute to fitness outcomes. We have now noted that during the life history cycles the gene-environment interaction is complex and dynamic. We further agree that though these factors should be recognized, their effect may be smaller compared to the specific extreme weather events we are examining. Therefore, it is possible that the strong/extreme events are masking the effect of other factors. Nonetheless we think it is safe to assume that these effects are minimal, and the control conditions we studied should adequately account for unknown carryover effects. We have also acknowledged that for future, more generalized situations, our specific scenarios have provided a valuable context to understand stress response. This is now included in the Discussion (Line 465-473).

From a scan of all of the DEGs, I spotted that some heatshock genes were in the lists. This is pretty interesting and such stress proteins are within the scope of the ms but are totally missing from the discussion. Specifically, HSPB9 was downregulated in the heart tissue between life history stages and upregulated in the snowstorm-induced stress response during incubation. The same gene was also downregulated in the liver when tested for the extreme spring. Another heat-shock gene only expressed in the liver tissue was HSPBAP1, downregulated both between the life history stages and the extreme spring arrival condition. Finally, HSPA5 that was downregulated in the heart during the incubation-snowstorm scenario is also very interesting. HSPA5 is known to regulate ER homeostasis (<https://doi.org/10.1111/brv.12667>) and has its upregulation in lean individuals has been suggested as a response to the physiological challenges that migratory birds face (<https://doi.org/10.1007/s00360-023-01529-x>). Overall, heat shocks seem to play a major role in stress response (<https://doi.org/10.1046/j.1461-0248.2003.00528.x>) and therefore it could be good to add a few statements in the discussion. there are many instances where such a discussion could be accommodated, e.g. L.376 where you almost touch the subject by stating that FKBP5 is a co-chaperone associated with Hsp90. Or in L.419 as another possible stress response mechanism. Or at least in L.445 as further genes of interest.

R5: We really appreciate the suggestions on the heatshock protein (HSP) gene family, and we agree that this gene family is of special interest, and acknowledge our oversight in discussion of these genes. We have included further discussion of the HSP gene family in Lines 205-210, 248-249, 279-282, 501-512.

L.375 ...and is suggested to be central in the regulation of HPA (<https://doi.org/10.1016/j.yhbeh.2021.105038>). Or this reference could work better in L. 424-425.

R6: We appreciate the suggestion. We have modified the text and the references.

Methods

While the study draws strong conclusions about the role of specific genes (mainly about FKBP5) in response to stress, it does not sufficiently explore potential relationships between genes. For example, I understand that you chose to construct a gene network for FKBP5 since it consistently popped up as differentially expressed and was the most up-regulated gene in the hypothalamus, but maybe a Weighted correlation network analysis would be more appropriate and provide more insight on the roles of different genes?

R7: We agree that a network analysis between all DEGs will provide more insight for the study. In addition, we also performed a WGCNA cluster analysis. However, due to the fact that there is no consensus in expression patterns across tissues, the biological heterogeneity was seen in the nature of this dataset. We have implemented the correlation between the modules of genes and the available phenotype measurements in Supplementary **Figure S11**. This should complement the Ingenuity Pathway Analysis (IPA) results to reveal biological functions pertaining to the DEGs shown in Supplementary Figure S6-S7. For instance, Fig.S6 shows the significant network or pathways enriched.

L536 blood plasma was also collected and stored till assay. But what assay? Nothing is mentioned about running any blood samples. I suppose that this is not needed here?

R8: Blood plasma was collected and corticosterone levels measured using radioimmunoassay. We now include the corticosterone levels as a phenotype. The method for corticosterone measurement is included in Line 603-607: Corticosterone concentrations were measured by radioimmunoassay as described by Wingfield et al. (1992) and Krause et al. (2015)^{62,63}.

L.617 The mitogenome was also assembled, the access of which should also be documented. In the interest of reproducibility, it is advisable to create a GitHub (or similar) page containing the commands and parameters used and the entire pipeline used for genome assembly and RNA-seq/DEG analysis. Some values are mentioned (e.g. L561) but maybe keeping them in a single place will facilitate transparency and accessibility for future researchers.

R9: The mitogenome sequence is incorporated in the genome assembly. We now highlight the accession number for the mitogenome, GenBank accession: genome assembly - GCA_039654755.1, MT sequence- CM077482.1. Additional scripts used to generate results and figures are also now available in the GitHub repository: https://github.com/wzuhou/LALO_scripts. A graphical pipeline can also be found in the same GitHub page.

Fig.2: Maybe a snail plot would be a more efficient way to describe/summarise the assembly (e.g. <https://doi.org/10.12688/wellcomeopenres.15679.1>). This way you can also report BUSCO results and scaffold statistics, so Table 1 can be moved to the supplement making the visualisation of your results more clear. Maybe panel C can be moved to the supplement too.

R10: We agree that a snail plot is a good way of visualizing the genome statistics, though we believe a table can convey more detail and exact statistics, thus we have included a snail plot in **Figure S2**.

Reviewer #2 (Remarks to the Author):

The present manuscript by Wu and colleagues aimed first to assemble the Lapland longspur genome, and then to determine the effect of environmental stressor (snowstorm) at different physiological stages (upon arrival and breeding). Several tissues were collected and subjected to RNAseq which allows the identification of many DEGs.

Although the study is very interesting and difficult (capturing wild birds, etc.), several concerns need to be addressed:

- The study is very descriptive.

R1: We appreciate the suggestions and comments from the reviewer, and fully acknowledge the concerns regarding the descriptive nature of the manuscript. However, herein we present the first genome assembly for the Lapland longspur, providing an important genomic resource for the research community to better understand avian genome biology, particularly in response to climatic changes. We also explore two different stress-response strategies in response to naturally-occurring extreme weather events using this free-living bird as a proxy through a large-scale transcriptomic study. These findings contribute to our understanding of the interplay between changing environments and genomic regulation and provide important foundational knowledge for future studies, highlighting genes that can be examined in more functional studies.

- Without any phenotypic data (body weight, fat content, liver weight, stress indicator, reproductive measurement), the study is very limited and does not provide much except variation in transcriptomics.

R2: We appreciate the suggestion to include phenotypic data, and we have now included the available phenotypic measurements in **Fig. S5**. However, please note that the sample size presented might not statistically represent the entire cohort across different conditions, but we believe these measurements complement the transcriptomic study and provide additional context.

- Some birds were captured in 2013 and other in 2016. Where there any effect of year? Also what was the age of birds? There were 20 birds in total, how many birds per season and physiological status?

R3: We fully acknowledge that the effect of year is unavoidable in this dataset, although we included comparisons between benign life-history stages to establish a baseline for the different physiological states. The two extreme weather events were naturally occurring and inherently confounded with the year. We recognize the importance of accounting for the year effect, and previous studies have examined snowstorms across different years. In future research, it would be interesting to explore transcriptomic differences under similar adverse conditions across multiple years. We have now addressed the inevitable year effect in the Discussion. All birds were adults at the time of capture. For each combination of weather condition, life-history stage, and tissue, we sequenced three replicates to ensure statistical power. In total, we present 66 RNA-seq samples and, where possible, sequenced tissues from the same individuals.

- Birds were captured with seed-baited potter traps- this might induce additional stress. How have authors controlled for that?

R4: Seed baited potter traps did not induce additional stress. All birds were sedated with isoflurane and euthanized by rapid decapitation with 3 minutes of capture (3 min 20 s \pm 52 s) and this is reflected in the corticosterone levels; i.e., not elevated to those which are observed with capture restraint. Even if there were some effects, these would be controlled across all samples and are not likely to cause false signals when comparing the differentially expressed genes (DEGs) as all the birds were captured using the same protocol to ensure consistency

- Line 173, in what tissue FBXO32 was up regulated?

R5: We now clarify that *FBXO32* is upregulated in hypothalamus-LHS in Line 191.

- Line 249, *UCN3* is urocortin 3. It belongs to CRF family, but it not CRF.

R6: We appreciate the suggestions and have now corrected in Line 272-274: “The gene *UCN3*, which encodes urocortin 3 and belongs to the corticotropin releasing factor family, was also down-regulated in the hypothalamus”.

Reviewer #3

General comments:

This is a nicely conceived and very well-written study that I enjoyed reading. I agree with the authors that these data are in high demand to provide a mechanistic, molecular, framework informing on how birds acclimate to environmental adversity. Some aspects of your work may require further attention, however. I have outlined some general remarks below, which are followed by a short list of line-specific comments that mostly pertain to style. I hope you will find these remarks useful as you prepare a revision.

1. The ms, valuable as it is, comes across as somewhat anecdotal as you draw on two case studies to make very affirmative conclusions about the molecular responses to extreme weather events and energy stress. It may well be that you are correct, but without replication there is really no way for us to gauge the reproducibility and generality of your data. I think you need to formally acknowledge this in the ms.

R1: We really appreciate the kind words and valuable comments from the reviewer, and have fully acknowledged the concerns, please see our detailed response to each question.

We agree with the reviewer that the two extreme weather events studied are unique cases, and cannot represent other extreme weather events and their impacts, but we believe the molecular investigation of these specific scenarios provides valuable context and insights for future studies on more generalized situations. We have also recognized other factors (such as carry over effect during migration) that may contribute to the response strategies and outcomes of wild-living animals. We have now included this point in the Discussion (Line 465-473).

2. While I do not mind the snowstorm example, I struggle more with the “late spring” case study where you contrast an adverse spring with a normal spring occurring three years later. This goes back to the issue of replication, because your study design inherently confounds year and weather event. It would be useful with additional data demonstrating similarly of gene expression profiles across a range of normal springs.

R2: We agree that the year effect is unavoidable in this dataset. The two extreme weather events were naturally occurring and inherently confounded with the year. We recognize the importance of accounting for the year effect, but we have previously reported the effects of extreme weather events on the HPA axis and physiology in arctic breeding songbirds across different years (Krause et al. 2018) (manuscript reference 7). In the future it would be interesting to explore transcriptomic differences under similar adverse conditions across multiple years. We have now included some text to discuss the confound of year effect in the discussion.

3. Your ms nicely sets the stage for transcriptional changes used to counter environmental perturbations and highlight two case studies answering to this. However, the seasonal progression result appears

somewhat disparate in this regard and I was not convinced that it fits into the overall narrative. It has great value still, but to really push the points you make in the Introduction, I advise you consider packaging the seasonal comparison into the online supplements. I understand that this could be useful for understanding the changes occurring in response to environmental adversity, but with the current packaging this does not really come through that clearly even after having read your discussion on the matter.

R3: We understand the reviewer's concern regarding the analysis of seasonal progression, and that the LHS comparisons are slightly different from the objectives. However, we believe gene expression profiles between life history stages under normal conditions set the baseline or background for physiological changes under extreme conditions. For instance, this comparison shows the changes of the ZP3 gene observed during the normal breeding cycle in the testes, highlighting its vital molecular function. Thus, when ZP3 is down-regulated in the extreme-spring, it is in line with the curtailed reproductive activity that was observed in the field. We have restructured the manuscript to emphasize this.

Line-specific comments:

L35: To make this point, you may want to defined extreme weather conditions, since many would argue that "yes, such are common, but are they really that common?".

R4: We have now included text about the detrimental weather conditions and the world climate reference in Line 35-41.

L35-38: Unclear, please revise.

R5: We have now rephrased this sentence.

L49: I agree and disagree. This is absolutely true for a cold challenge, but in the context of heat stress, water may become limiting much earlier than energy ever becomes an issue.

R6: I agree that other potential factors can contribute to the allostatic load, and the overall energetic supply should include resources such as water and food. We have rephrased this sentence.

L58-59: I advise you unpack this a bit more to set the stage for your study.

R7: We have rephrased this sentence.

L79: This will sound minor to any non-specialist and so I suggest you provide more biological context of this delay.

R8: Thanks for the suggestions, we have modified in Line 84-86.

L97, L111, L132 (- L146): I would place these three paragraphs in the Methods, because they do not really answer to your research problem.

R9: We have modified the overall text to focus more on the research question. But the high-quality genome assembly provides important genomic resources for future study, and we believe is important to present this foundation for readers.

L346-355: This is largely repetitive of the Introduction and can be removed.

R10: We have now removed the repetitive paragraph.